# Inka Child Mummy Found in Cerro Aconcagua (Argentina) Traced Back to Populations of the Northern Peruvian Coast through Y-Chromosome Analysis

José R. Sandoval [1,*], Ricardo Fujita [1], Marilza S. Jota [2], Thomaz Pinotti [2,3] and Fabrício R. Santos [2,*]

[1] Centro de Investigación de Genética y Biología Molecular, Instituto de Investigación, Facultad de Medicina Humana, Universidad de San Martín de Porres, Lima 12, Peru; rfujitaa@usmp.pe

[2] Laboratório de Biodiversidade e Evolução Molecular, Instituto de Ciências Biológicas, Universidade Federal de Minas Gerais, Belo Horizonte 31270-010, MG, Brazil; marilza.sileia@gmail.com (M.S.J.); thomaz.pinotti@gmail.com (T.P.)

[3] Centre for GeoGenetics, University of Copenhagen, 1350 Copenhagen, Denmark

\* Correspondence: jsandovals@usmp.pe (J.R.S.); fsantos@icb.ufmg.br (F.R.S.)

**Abstract:** The mummy of a seven-year-old child that was discovered in 1985 in Cerro Aconcagua (Mendoza, Argentina) was likely part of an Inka sacrificial religious practice known as *capacocha*. Previous uniparental DNA marker studies conducted by some scholars have suggested that the mummified child may be related to the southern Andean population of Peru. However, autosome genome-wide analysis performed by others has indicated that the child was more closely related to the population along the northern Peruvian coast than to that of the southern Andes. In this study, we aimed to determine possible genealogical connections in the male lineage of the mummified child. To achieve this, we compared the genetic profile of the mummy with an extensive database of contemporary individuals from the northern Peruvian coastal and southern Andean regions. We used single nucleotide polymorphisms and short tandem repeats from the nonrecombining region of the Y-chromosome for our analysis. Our results confirmed that the Inka child mummy was closely related to individuals from the north coast of Peru. This suggests that the child was likely descended from the Muchik–Chimor-speaking people.

**Keywords:** Y-chromosome; Inka child mummy; Aconcagua; Peru

## 1. Introduction

Child sacrifices (*capacocha*) were a central feature of Inka religious ceremonies, and according to contemporary chronists, they were conducted at *waka* (or *huaca*) sacred sites and governed by a specific calendar [1]. Most *waka* were on mountains and volcanoes and were considered *Apu* deities. In 1985, a team of hikers discovered the remains of a mummified seven-year-old child in Cerro Aconcagua, located in Mendoza, Argentina. The Aconcagua is the highest mountain in the Americas, at 6960.8 m.a.s.l., and the Inka child mummy was found at about 5300 m.a.s.l. The mummy is currently preserved at the Instituto de Ciencias Humanas, Sociales y Ambientales, Incihusa-Conicet, CCT Mendoza (Argentina). The textile remains are conserved at the Museo Arqueológico from FFyL of the Universidad Nacional de Cuyo (UNCuyo).

The child was likely sacrificed during the pre-Columbian period, approximately 500 years ago [2,3]. As was typical for Inka child sacrifices, the boy was buried with a rich funerary context. This included a *cumbi*, a luxury textile with pelican iconography; six statuettes (three made from *Spondylus* sp.) with tropical bird feathers (macaw, toucan); a *chuspa* bag with coca leaves; a necklace made of *Spondylus* pieces; and other manufactured articles. Bioarcheological analysis has revealed that his *unku* tunic was stained with vomit remains containing a red pigment of achiote (*Bixa orellana*) [4,5]. But what about his genetic affiliation?

From paternal DNA genealogical studies, it is well known that almost all Native American populations descend mainly from a major common founder, known as the Q-M3 haplogroup, followed by some minor lineages (e.g., Q-CTS1780) and two rare subtypes of C lineages [6]. To complement this binary polymorphism (single nucleotide polymorphism), short tandem repeats (STRs) of the nonrecombing part of the Y-chromosome (Y-STRs) are widely used for paternity testing, familial genealogy, forensic analysis, and kinship identification. The dual combination of Y-chromosome single nucleotide polymorphism (SNP) and Y-STR analyses allows scholars to refine the genealogies within or between human population groups. Due to the advantages of these Y-chromosome markers, the child mummy's DNA has been subjected to investigation in a similar way to what was performed with mitochondrial DNA (mtDNA) [7,8]. Regarding this, by maternal line, in the autochthonous South American population, four typical founder haplogroups were found, A2, B2, C1, and D1/D4h3, whose frequencies vary depending on the geography and demographic history of a subpopulation, although B2 is a prevalent lineage in the Central Andes.

Previous genetic investigations have indicated that the Inka child of Aconcagua descended from Native Americans bearing the haplogroups C1b and Q-M3 from their maternal (mtDNA) and paternal (Y-chromosome) lineages, respectively [7,8]. Both lineages have a pan-American distribution and are believed to have originated in Beringia more than 15,000 years ago [6,9,10]. A preliminary analysis of these uniparental DNA markers suggested that this mummy may be related to individuals from the southern Andes region of Peru [7,8]. However, a recent paleogenetic study [11] compared the ancient DNA with contemporary data from the Central Andes and found that the genomic profile of the child mummy was more closely related to the population from the northern Peruvian coast than to that of the Andean region. Specifically, the study found similarities with the ancient DNA from the El Brujo archaeological complex (1750–560 years before present (ybp)), a ceremonial center associated with Muchik populations (1870–1170 ybp) [11–13].

Most studies of the coastal Peruvian population have focused on the broader archaeological and linguistic context, and few have concentrated on the genetic relationships with the Inka child mummy [11,12]. According to archaeological and anthropological records, on the northern Peruvian coast, the Muchik and Chimor cultures were two of the earliest and emergent societies in the pre-Columbian period. Before the Inka conquest, the northern coastal population extended its territories along the Pacific coast, mainly from Piura to the coast in the Ancash department. Furthermore, some groups of artisans, such as potters inhabiting the Piura Valley and nearby sites, were called Northern Muchiks since they were associated with the Muchik culture of Lambayeque [13]. On the other hand, on the southern coast, Nazca and Paracas cultures were followed by the emergence of the Wari culture. In the Southern Andes and Altiplano, the Tiwanaku flourished until, finally, the Inka Empire. The pre-Inkas and the Inka cultures are collectively known as Central Andes civilizations. Strikingly, genetic studies of ancient DNA versus that of contemporary populations recapitulated this geographic map [11]. Thus, there is a genetic homogeneity mainly along the Central Andes (which includes the coast), encompassing areas of what was the Inka Empire, although some populations have been differentiated due to genetic drift or low gene flow [14]. Most populations across the Central–Southern Andes of Peru, including western Bolivia and northern Chile, show similar genetic profiles among them, indicating a genetic continuity [11]. Considering this historical and genetic makeup of the Peruvian population, under the lens of DNA, any genetic signature can be traced, at least to a specific geographic region.

To investigate the potential genetic links of the child mummy's paternal lineage, we conducted a study searching the Peruvian Y-chromosome database (contemporary Andean and northern Peruvian coastal populations) and compared it with the genetic profile of the child mummy [3,8]. Thus, to answer this genealogical question, we utilized SNPs and STRs from the nonrecombing region of the Y-chromosome. We found consistency with previous autosomal ancient DNA studies.

## 2. Materials and Methods

### 2.1. Ethics Statement and Sampling

Buccal swab sampling of Indigenous participants was conducted after providing the partici­pants with details regarding the project and obtaining signed informed consent. The expeditions were conducted in 2009 and 2014, and relatives were avoided during the sampling process. This study was approved by the Ethics Committee of the University of San Martín de Porres (Oficio No. 818–CIEI-USMP-CCM, 2011; Peru). The study and protocols were conducted in accordance with the ethical recommendations of the Declaration of Helsinki. The comprehensive sampling included the published Y-chromosome data from all native Peruvian individuals in our database [15–19]. Thus, for a more detailed analysis, we selected 110 samples from northern Peru, including 52 from Piura (Sechura = 9, Tallan = 3, Narihuala = 7, Catacaos = 6, La Union = 1, Piura = 8, Chulucanas = 9, La Arena = 4, La Matanza = 1, Tambo Grande = 2, Morropon = 1, and Tablazo = 1), 43 from Lambayeque (Chotuna = 15, Lambayeque = 17, and San Jose = 11), and 11 from La Libertad (Huanchaco = 11). Additionally, we included four samples from Cajamarca based on the data obtained from Sandoval et al. [15,18]. The details and geographic coordinates of the Indigenous communities sampled are presented in Supplementary Table S1, and a map of the approximate geographical locations is in Figure 1a.

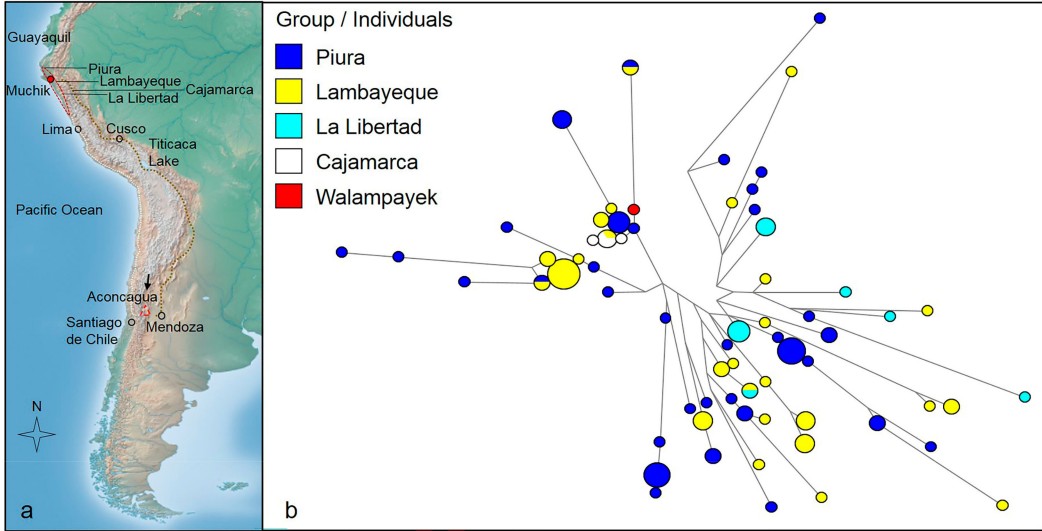

**Figure 1.** Map highlighting the locations discussed in the text and the phylogenetic network. (**a**) Nucleus of Muchik culture expansion in Peru (150–850 AD) and Aconcagua Mountain (Mendoza, Argentina). The red point indicates the most probable geographic affiliation of Walampayek to be the northern Peruvian coast. The dotted lines indicate the possible routes of the Khapaq Ñan used by the Inkas to reach the Aconcagua. (**b**) Median-joining network for the 17 Y-STR haplotypes in Peruvians depicted by different colors. Haplotypes are represented by circles and size according to the number of individuals, and branch lengths are proportional to the number of mutation steps. The Inka child mummy was called Walampayek by us (in red).

### 2.2. Genotyping the Y-Chromosome SNPs and STRs

Genomic DNA samples were previously genotyped for Y-SNPs, primarily Q-M242, Q-M3, and Q-M346, using TaqMan assays [18]. This was performed following the stan­dardized TaqMan protocol and using the genotyping universal polymerase chain reaction (PCR) master mix (2×), TaqMan genotyping assay (20×), labeled DNA probes (FAM and VIC), and the DNA sample to be tested. For a volume of 2.75 µL, the master mix reaction consisted of 2.5 µL TaqMan Genotyping Master Mix and 0.25 µL TaqMan Genotyping Assay Mix. Thus, 2.75 µL master reaction was transferred to each well in a 96-well optical PCR plate, to which a volume of 2.25 µL DNA sample (1–10 ng) was added. For 50 PCR cycling conditions, the temperatures were 95 °C for 15 s and 60 °C for 1.5 min. The genotyping of

the samples was determined using the SDS software version 2.2 (AutoCaller) in the RT-PCR 7900HT system (Thermo Fisher Scientific, Waltham, MA, USA).

Deep sequencing of the human Y-chromosome was performed on some samples, and the data were published in a previous study [6]. Selected samples were subsequently genotyped using the Y-filer PCR amplification kit, which included 17 Y-STR markers located in the nonrecombining region of the Y-chromosome. These markers were DYS19, DYS385a, DYS385b, DYS389a, DYS389b, DYS390, DYS391, DYS392, DYS393, DYS437, DYS438, DYS439, DYS448, DYS456, DYS458, DYS635, and YGATAH4.

Genotyping of the Y-STR amplicons was performed using the ABI3130xl Genetic Analyzer (Thermo Fisher Scientific, Waltham, MA, USA), and the STR fragments were sized utilizing GeneMapper v3.2 software (Thermo Fisher Scientific). The PCR multiplex consisted of 2.3 µL reaction mix, 1.25 µL primer set, 0.2 µL AmpliTaq Gold, and the addition of 1.5 µL $H_2O$ milli-Q. For the PCR reaction, the temperatures were 94 °C for 1 min, 61 °C for 1 min, and 72 °C for 1 min (30 cycles).

Genotypic data, including the Y-SNPs and Y-STRs, of the Inka child mummy were previously reported by Salas et al. [8] and Moreno-Mayar et al. [3].

### 2.3. Y-STR Data Analysis

The DYS389b alleles were scored by subtracting DYS389I from DYS389II. To determine genetic relationships at the individual level, we utilized the median-joining and maximum parsimony algorithms [20,21] present in Network 10.2.0.0 phylogenetic software (www.fluxus-engineering.com) (accessed on 27 February 2023). The weight criteria for the Y-STRs were determined according to Sandoval et al. [15]. We utilized the ρ (rho) statistic described by Forster et al. [22] to estimate the time to the most recent common ancestor (TMRCA). An average effective mutation rate of $6.9 \times 10^{-4}$/locus/25 years was used for the calculations, as reported by Zhivotovsky et al. [23]. To confirm the Y-haplogroup assignment of the samples with missing Y-SNP data, we utilized the Bayesian Whit Athey haplogroup predictor [24] based on the Y-STR data.

To explore the Y-STR haplotype profiles, as well as visualize pairwise differentiation at the individual and the population levels, we used principal components analysis (PCA) implemented in the FactoMineR v. 2.8 package and available in the R project (http://www.r-project.org) (accessed on 15 August 2023).

### 2.4. Y-SNP Data Analysis

To refine whether the Aconcagua child mummy belonged to a previously undescribed Y-chromosome subhaplogroup with other individuals from the northern Peruvian coast, we compared it with the data from four ancient DNA reports by Bongers et al. [12] and Nakatsuka et al. [11] with sufficient coverage. We used bcftools mpileup and call functions [25] to obtain the genotypes in the 10 Mb accessible region of the Y-chromosome [26] and excluded the indels, triallelic positions, alleles not found in >90% of the population of nonclonal reads at a locus, and C-to-T and G-to-A transitions in the forward and reverse reads, respectively. We determined the haplogroups by matching the ancestral and derived variants with both the ISOGG 2019–2020 (https://isogg.org/tree/) (accessed on 17 May 2023) and curated SNP list from Pinotti et al. [6], which includes private markers.

## 3. Results
### 3.1. Y-STR Results

The low-coverage genome of the Inka child mummy was previously published by Moreno-Mayar et al. [3]. Its Y-chromosome lineage was defined by a few informative SNPs and belongs to the Q-M3 haplogroup branch [8], which is known to reflect the population expansion that began approximately 15,000 years ago [6] but allows for no finer resolution. Therefore, a more detailed phylogenetic analysis was performed using Y-STRs.

We conducted a comprehensive search of the Y-STR Peruvian database using the genetic profile of the Inka child mummy and selected 110 individuals from the northern

Peruvian populations for the reconstruction of a phylogenetic network (Figure 1b). This was performed in consideration of a previous report by Nakatsuka et al. [11], as well as the closest genetic relationship of the child mummy that had been found in the comprehensive search, which included all genotyped Peruvian samples [15–19].

The Y-STR phylogenetic network revealed a close genetic relationship between the mummified child (Walampayek) and individuals from Piura, Lambayeque, and Cajamarca (Figure 1b, Table 1). A single mutation separated the Y-STR haplotype of Walampayek from that of an individual from Catacaos, Piura (sample MAL10). MAL10 also differed from another haplotype that was shared by individuals from Piura, including Catacaos (sample VICUS7), La Arena (sample VICUS4), Narihuala (sample NARI6), and Chulucanas (sample VICUS2), by a single mutation step. In addition, the MAL10 sample was also differentiated from the haplotype of a Cajamarca individual (WASI36) by a one-step mutation, showing a close affinity with individuals from Lambayeque (POMAC-12) and Cajamarca (CAX25, Q_Caj15). Similarly, an individual from Chotuna, Lambayeque (sample CHO-12), differed by a one-step mutation from the shared haplotype of VICUS2, VICUS4, VICUS7, and NARI6. Moreover, two related haplotypes, one each from Lambayeque (Pomac02 and Pomac05) and from Cajamarca (Q_Caj17), were included in this cluster. With respect to the other haplotypes, most were closely related to each other. Two shared haplotypes were observed between individuals from Piura and Lambayeque departments. Apart from one shared haplotype between two individuals from Lambayeque and La Libertad, other haplotypes from individuals inhabiting La Libertad were scattered along the divergent branches in the reconstructed phylogenetic network.

**Table 1.** The closest haplotypes among the individuals using 17 Y-STR markers.

| Sample/Locality | 17 Y-STR DYS19-DYS385a-DYS385b-DYS389a-DYS389b-DYS390-DYS391-DYS392-DYS393-DYS437-DYS438-DYS439-DYS448-DYS456-DYS458-DYS635-YGATAH4 | Reference |
|---|---|---|
| Child mummy of Aconcagua (Mendoza, Argentina) | 14-16-16-14-17-25-10-14-13-14-11-12-20-15-16-22-12 | Salas et al. [8] |
| MAL10 (Catacaos, Piura) | 14-15-16-14-17-25-10-14-13-14-11-12-20-15-16-22-12 | This study |
| VICUS2, VICUS4, VICUS7, NARI6 (Chulucanas, La Arena, Catacaos, Narihuala; Piura) | 14-14-16-14-17-25-10-14-13-14-11-12-20-15-16-22-12 | This study |
| WASI36 (Cajamarca) | 14-15-15-14-17-25-10-14-13-14-11-12-20-15-16-22-12 | Sandoval et al. [18] |
| CHO-12 (Chotuna, Lambayeque) | 14-14-16-14-17-25-10-14-13-14-12-12-20-15-16-22-12 | This study |
| Pomac02, Pomac05 (Lambayeque) | 14-14-16-14-17-25-10-14-13-14-11-12-20-15-15-22-12 | This study |
| POMAC-12, Q_Caj15, CAX25 (Lambayeque, Cajamarca) | 14-14-15-14-17-25-10-14-13-14-11-12-20-15-16-22-12 | This study, Sandoval et al. [15,18] |
| Q_Caj17 (Cajamarca) | 14-14-15-14-17-25-10-14-14-14-11-12-20-15-16-22-12 | Sandoval et al. [15] |

To estimate the date of TMRCA of the cluster (to which Walampayek was connected), we assumed the representative modal haplotype (that of VICUS2, VICUS4, VICUS7, and NARI6) to be the "ancestral haplotype" and considered the other related haplotypes, including that of the child mummy, its descendants. The results showed that TMRCA was estimated to have lived approximately 1978 years ago (with a standard deviation of ±874 years).

### 3.2. Y-SNP Results

We found that the child mummy of Aconcagua was missing all positions that define the Q-M848 subhaplogroup but was positive (one read support for a transversion SNP) for the downstream marker MPB073. This subhaplogroup, however, was ~14,000 years old and only included two individuals from Lima and an Inka descendant from Cusco [6],

and it was, therefore, not especially informative in establishing the other relationships of the child mummy. Individuals with sufficient coverage from the El Brujo archaeological complex belonged to common coastal Andean sublineages (Q-CTS4000 and Q-Y788, both downstream of Q-M848), and no higher resolution than Q-CTS1780 and Q-M848 was possible for individuals from the Chincha Valley of southern Peru. The SNPs defining the haplogroup of ancient DNA are reported in Supplementary Table S2.

### 3.3. Principal Component Analysis Results

For the Y-STR haplotype comparisons, we used previously collected samples, including published data and the child mummy haplotype, from different populations from Peru, Bolivia, Ecuador, and Brazil that were part of the South American Genographic project. After plotting Q-M3 Y-STR haplotypes, 632 individuals were selected for visualization in the bidimensional space. The Peruvian population database [15–19,27] consisted of the following conventional macroregions: Coast North (*n* = 105, from Piura, Lambayeque, and La Libertad) and Andes South: Cusco (*n* = 65); Chivay, Arequipa (*n* = 17); Chanka, Apurimac (*n* = 17); Huancavelica (*n* = 42); Chopcca, Huancavelica (*n* = 21); Altiplano–Peru and Bolivia (*n* = 154); Andes North: Cajamarca (*n* = 22); Kañaris and Inkawasi, Lambayeque (*n* = 54); Ancash (*n* = 4); Amazonia: Jivaroan, from San Martín and Loreto (*n* = 39); Ashaninka from Satipo, Junín (*n* = 16); and Andes Ecuador (*n* = 76, Quichua, Karanki, Cañar, and Pastos).

The PCA scatter plot showed the child mummy Y-STR haplotype to be linked to the northern Peruvian population (Figure 2). Specifically, the Walampayek haplotype was closely associated with some coastal inhabitants from Piura and Lambayeque departments (as shown in Figure 1b), and this observation recapitulates the panorama on clinal genetic distance relationships in the Central Andes. Thus, the northern Peruvian population is more closely related to the Andean population from Ecuador than to the northern Peruvian Amazonia population, such as the Jivaroan people, whereas Altiplano (Peru and Bolivia, including Lake Titicaca) and southern Peruvian Andes populations are closely related to each other. Despite the fact that most people who speak Ashaninka (an Arawakan language group) are differentiated from the coast as well as the Andes, they have more genetic affinity with Jivaroan-speaking people than with the rest. On the other hand, increasing the number of samples (from the Y-STR database and unpublished data) would not change the observed scenario. When we explored the Y-STR haplotype of the child mummy in the Amazonian population of Peru, Bolivia, Ecuador, and Brazil (South American Genographic project database and published data) [19,27,28], no direct genealogical relationship was found. As we have shown above, the child mummy Y-STR haplotype was linked only to the northern Peruvian population.

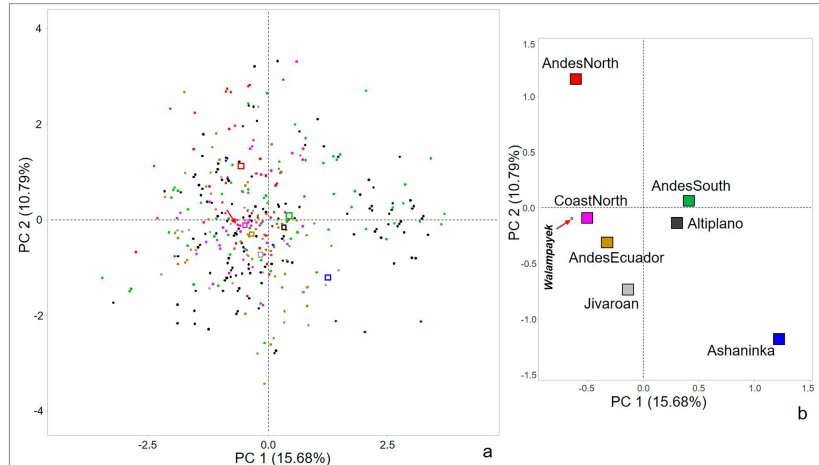

**Figure 2.** PCA scatter plot for 17 Y-STR haplotypes (*n* = 633, including Walampayek). (**a**) Relationships at the individual level. (**b**) Relationships at the population level. Centroids are represented by squares, and the child mummy haplotype is indicated by a red arrow.

## 4. Discussion

Before and during the Inka Empire, child sacrifices were characterized by ritual offerings in a special ceremonial place, and they were probably dedicated to local deities, idols, or fertility. This scenario may have been the case for the Inka child mummy and the other mummified children's bodies found on volcanoes Ampato (6290 m.a.s.l., Arequipa), Sara Sara (5505 m.a.s.l., Ayacucho), Picchu Picchu (5664 m.a.s.l., Arequipa), Misti (5820 m.a.s.l., Arequipa), and Llullaillaco (6739 m.a.s.l., Argentina) [2,29,30]. In contrast, in more recent years, on the north coast of Peru, archaeologists found massive remains of sacrificed children (in the Huanchaquito–Las Llamas site, Moche Valley) that were associated with the Chimor culture (1450 AD) [31]. Since DNA data on some of them are available [8,11,32], we can address the question of the genetic ancestry of the Inka child mummy.

Autosome genome-wide analysis of the ancestry of the Inka child mummy found at Cerro Aconcagua (Mendoza, Argentina) indicated that the northern Peruvian coast was likely his geographic affiliation [11,12]. However, previous studies considering his uniparental markers have suggested that there was a southern Andean affinity instead [7,8]. Probably due to the absence of published Y-STR data on contemporary northern coastal Peruvian populations, Salas et al. [8] did not detect the genetic relationship with the child mummy, as did other works using autosomal aDNA studies [11,12]. However, the inclusion of the Cajamarca population data (Q_Caj15 and Q_Caj17 samples from Sandoval et al. [15]) could have changed their interpretations. These studies show us how, in some cases, a lack of population data may lead to different conclusions or interpretations.

According to historical records, Emperor Tupac Yupanqui (as documented by Cabello de Balboa [33]) conquered several coastal and Andean communities, mostly descendants of the Muchik and Chimor peoples, during the Inka expansion in northern Peru. Some Muchik families, particularly those from the Jayanca Valley, were relocated as *mitmakunas* to various regions of the Tawantinsuyu, including Trujillo and Huamachuco (La Libertad), Cajamarca and Tallan (Piura), Huacho (north of Lima), Chincha and Cañete (Ica), and Cusco and Copacabana (Titicaca Basin) [12,34]. In the case of the Inka child mummy, although the exact interment date is unknown, it is likely that he was transported to Cusco via the Khapaq Ñan routes. Following ritual ceremonies, he was laid to rest in Cerro Aconcagua in Argentina (Figure 1a). After observing the peculiarities of the *capacocha* ritual that involved the Inka child mummy, some scholars were led to suggest a close association with the pre-Columbian Peruvian coastal cultures [2,4].

In the present study, we utilized Y-chromosome data to identify the closest Y-STR profiles between the mummified child's haplotype and that of contemporary individuals from the northern Peruvian populations, specifically those from Piura, Lambayeque, and Cajamarca. Assuming that the individuals WASI36 and Q_Caj17 are directly related to individual POMAC-12 from Lambayeque (who shares a haplotype with CAX25 and Q_Caj15), then the four individuals from Cajamarca could be descendants of Piura or Lambayeque (given that they are most closely related to each other and to the Piura and Lambayeque subpopulations; Figure 1b). This supports the idea of gene flow between the coastal and Andean regions. In addition, males from the Tumbes, Piura, Lambayeque, and Chachapoya subpopulations share recent common ancestors, as reported by Barbieri et al. [14]. Piura and Lambayeque have strong historical links with each other, and archaeological evidence has shown demographic occupation by Muchik, Vicus, and Tallan cultures since distant times [13]. Furthermore, the communities of La Arena, Narihuala, and Catacaos are in the Piura Valley, near Chulucanas. These four localities are connected to the Lambayeque region, where the Muchik culture flourished, through the Jayanca Valley. The Muchik culture also encompassed the Huarmey Valley in the area of the Ancash department [13]. Within this context, Lambayeque and La Libertad were also connected to the Cajamarca culture [13]. The estimated time to MRCA for this cluster of closely related Y-STR haplotypes, including the child mummy (Walampayek), suggests an early period of expansion for their founders on the northern coast of Peru. This ancestor likely lived approximately 2000 years ago. Perhaps some Muchik individuals from the Muchik group

at the El Brujo archaeological complex in La Libertad [11] are genetically linked to these ancient patriarchs. Although the estimated TMRCA was inferred through phylogenetics, it should be interpreted with caution. TMRCA can vary depending on the mutation rates and number and types of markers evaluated.

In addition, when we chose only 15 Y-STRs (without including the DYS385 marker) to explore the child mummy haplotype from the Peruvian population database (South American Genographic project, and from other data), a shared haplotype was observed between Walampayek and some individuals from Piura (VICUS2, VICUS4, VICUS7, NARI6, and MAL10 samples); Lambayeque (POMAC-12 sample); Cajamarca (Q_Caj15 sample from Sandoval et al. [15]; CAX25 and WASI36 samples from Sandoval et al. [18]); Kañaris locality, Lambayeque (KAN7 sample from Sandoval et al. [18]); and a Jivaroan speaker (CHP-352 sample from Guevara et al. [35]). However, this last one was differentiated by four-step mutations from Walampayek when compared with 23 Y-STRs. Moreover, the haplotype of the CHP-352 individual was very differentiated from Jivaroan-speaking people. This self-identified individual was likely sampled in the Chachapoyas city [35], and as an outlier, this Jivaroan was not included in the analyses. However, it could be interpreted as an occurrence of gene flow from the North Coast/Andes to a Jivaroan community (northwestern Amazonia, Amazonas department). Nevertheless, there might be another interpretation.

Furthermore, a separate study conducted on the Ecuadorian population revealed a genetic similarity (albeit distinguished by four mutation steps) between an individual from Guayaquil, Province of Guayas (sample ID P27209P), an Ecuador coastal region [36], and Peruvians from Piura (as evidenced by the shared haplotype between VICUS2, VICUS4, VICUS7, and NARI6). This connection could reflect either common ancestry or gene flow that occurred between the northern Peruvian and Ecuadorian populations due to increased trade networks since pre-Inka times [13]. Iconographic representations have also demonstrated the intercultural connections between the Muchik (coastal), Cajamarca (Andes), and Jivaroan (Amazon) peoples [37]. This may explain why populations along the northern Peruvian coast (e.g., Tumbes, Tallan, Narihuala, Chulucanas, Chotuna, and Eten) currently share extended blocks of ancestry profiles with nearby Andean populations [14]. Overall, this panorama is consistent with the clinal pattern of genetic relationships by distance [14] and supports the genetic affiliation of the Inka child mummy with the northern coast of Peru, as reported by Nakatsuka et al. [11] and Bongers et al. [12]. Furthermore, the genetic similarity between Walampayek and an individual from the Chotuna community, separated by only three mutation steps, may indicate a historical connection with that of the fabled builders of Chot, a ritual center likely referred to as *waka Chotuna* in the Muchik language, but now known as Chotuna. Alternatively, this genetic similarity could suggest a shared ancestry with the Lambayeque culture, as noted by Cabello de Balboa [33].

A recent investigation of ancient DNA from occupants of Machu Picchu and Cusco (Peru) showed that individuals from different ecological regions (coast, Andes, Amazonia) were transported by the Inkas as *mitmakunas*. These observations agree with the chronicles, which indicated that a high gene flow occurred along the Tawantinsuyu (four regions from Southern Colombia to the north of Chile, including Ecuador, Bolivia, and northwest Argentina). Likewise, this panorama has been shown by previous genetic studies [11,12,17,18]. Moreover, aDNA profile comparisons among Native Americans (from North America to South America) have shown the Inka child mummy's ancestry to also be linked to an individual from the Cajamarca department (Peru_Cajamarca_ISA1 sample; Figure S7) [38]. It is noteworthy that our inference was congruent with previous studies.

In conclusion, our analysis reveals that consistent with the autosome genome-wide studies of Walampayek [11,12], his male lineage suggests a stronger genetic affinity with contemporary individuals from the northern coast of Peru. This finding supports the hypothesis that Walampayek was likely a descendant of the Muchik–Chimor-speaking people.

**Supplementary Materials:** The following supporting information can be downloaded at: https://www.mdpi.com/article/10.3390/dna3040012/s1, Table S1: list of the 17 Y-STR haplotypes genotyped in the studied populations (*n* = 110), Table S2: SNPs defining the haplogroup of ancient DNA.

**Author Contributions:** J.R.S.: conceptualization, formal analysis, investigation, methodology, validation, visualization, resources, writing—original draft preparation, writing—review and editing. R.F.: resources, writing—review and editing. M.S.J.: formal analysis, writing—review and editing. T.P.: formal analysis, investigation, validation, writing—review and editing. F.R.S.: resources, writing—review and editing. All authors have read and agreed to the published version of the manuscript.

**Funding:** This research was funded by the National Geographic Society (The Genographic Project-South America, and Geno.2, entitled Origins and descend of the Inca Empire, grant number 5-13), USMP (E10012019010) of Peru, FAPEMIG, and CNPq of Brazil.

**Institutional Review Board Statement:** The study was conducted in accordance with the Declaration of Helsinki and approved by the Ethics Committee of the University of San Martín de Porres (Oficio N° 818-CIEI-USMP-CCM, 2011; Lima, Peru).

**Informed Consent Statement:** Informed consent was obtained from all subjects involved in the study.

**Data Availability Statement:** Supplementary Materials is available for this article.

**Acknowledgments:** We thank the participants who contributed their biological samples. We thank Carlos Wester of the Chotuna-Chornancap Site Museum, Lambayeque, Peru, for his support. A special thanks to Oscar Acosta for his support during fieldwork.

**Conflicts of Interest:** The authors declare no conflict of interest. The funders had no role in the design of the study; in the collection, analyses, or interpretation of data; in the writing of the manuscript; or in the decision to publish the results.

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
