# Peer review of "Inka Child Mummy Found in Cerro Aconcagua (Argentina) Traced Back to Populations of the Northern Peruvian Coast through Y-Chromosome Analysis"

_2673-8856, doi:10.3390/dna3040012_

Round 1

Author Response

The article brings together new genetic data from modern individuals residing in the northern Peruvian coastal and southern Andean areas. It aims to advance our understanding of the paternal ancestry of the Inka child mummy.

There are some inconsistencies between the pdf file and word format. It seems that “3.3. PCA results” section is missing from the word file. Therefore, I performed my assessment based on the pdf version. Please find my comments and suggestions below:

Abstract, lines 14-17: The authors should make clearer the fact that that these conclusions are drawn based on previous/other studies.

“The preliminary analysis of uniparental DNA markers suggested that the mummified child may be related to the southern Andean population of Peru. However, autosome genome-wide analysis indicated that the child was more closely related to the population along the northern Peruvian coast than to those of the southern Andes.”

We modified:

“Previous uniparental DNA marker studies conducted by some scholars have suggested that the mummified child may be related to the southern Andean population of Peru. However, autosome genome-wide analysis performed by others has indicated that the child was more closely related to the population along the northern Peruvian coast than to that of the southern Andes.”

Introduction, line 45: “On the other hand, by using single-nucleotide polymorphisms of the Ychromosome…” The shift to this next paragraph is rather abrupt and “on the other hand” doesn’t help much. Please consider rephrasing this part.

We connected the ideas with following:

“…But what about his genetic affiliation?”

“From paternal DNA genealogical studies, it is well-known that almost all Native American populations descend mainly from a major common founder, known as the Q-M3 haplogroup,…”

Introduction, lines 53-54: “Due to these advantages of the Y-chromosome markers, the child mummy DNA was subjected to investigation, in similar way that was done with mitochondrial DNA (mtDNA)”. The part regarding mtDNA is confusing: when was the mtDNA analysis performed; in this paper or others?

After editing, we added the references:

“Due to the advantages of these Y-chromosome markers, the child mummy’s DNA has been subjected to investigation in a similar way to what was done with mitochondrial DNA (mtDNA) [7,8]. ”

Introduction, lines 92-96: It is important to emphasize at this point that this study involved the construction of a modern DNA database for the purpose of comparing it with the previously reported genetic signature of the child mummy. This fact only becomes evident later in the text.

We rephrased:

“To investigate the potential genetic links of the child mummy’s paternal lineage, we conducted a study searching the Peruvian Y-chromosome database (contemporary Andean and northern Peruvian coastal populations) and compared it with the genetic profile of the child mummy [3,8]. Thus, to answer this genealogical question, we utilized SNPs and STRs from the non-recombining region of the Y-chromosome. We found consistency with previous autosomal ancient DNA studies.”

Introduction, line 176: “Therefore, a more detailed analysis was performed using Y-STRs.” What type of analysis (phylogenetic, phylogeographic, etc)?

It is done:

“Therefore, a more detailed phylogenetic analysis was performed using Y-STRs.”

Results, line 254: There is a “Figure not shown” which should be fixed.

It was deleted.

Results, lines 327-332: “phylogenetic network not shown” Why? If not highly relevant, maybe it can be included in the supplementary files. Please rephrase the following: “As an outlier individual, it was not included in the analyses, because of it appears to be a…”

We deleted “phylogenetic network not shown” due that we explained in the text how the searching was done in the database (published and unpublished data), and the finding of closest Y-STR haplotypes with that of the child mummy, but finally we opted by 17 Y-STRs for the phylogenetic reconstruction (Figure 1b phylogenetic network). Also, we rephrased those items as was suggested:

“In addition, when we chose only 15 Y-STRs (without including the DYS385 marker) to explore the child mummy haplotype from the Peruvian population database, a shared haplotype was observed between Walampayek and some individuals from Piura (VICUS2, VICUS4, VICUS7, NARI6, and MAL10 samples); Lambayeque (POMAC-12 sample); Cajamarca (Q_Caj15 sample from Sandoval et al. [15]; CAX25 and WASI36 samples from Sandoval et al. [18]); Kañaris locality, Lambayeque (KAN7 sample from Sandoval et al. [18]), and a Jivaroan speaker (CHP-352 sample from Guevara et al. [35]). However, this last one was differentiated by four step mutations from Walampayek, when compared with 23 Y-STRs. Also, the haplotype of the CHP-352 individual was very differentiated from Jivaroan-speaking people. This self-identified individual was likely sampled in the Chachapoyas city [35], and as an outlier, this Jivaroan was not included in the analyses. However, it could be interpreted as an occurrence of gene flow from the north Coast/Andes to a Jivaroan community (northwestern Amazonia, Amazonas department). Nevertheless, there might be another interpretation.”

Maybe it would worth to briefly state why different studies point to different genetic ancestries for this individual (different genomic markers with various resolutions show different parts of the story, the dimension of the comparative modern and ancient datasets and so on).

If Salas et al. (2018) had been merged into Cajamarca Y-STR data (northern Andes), they probably could be found the genetic link at least with Q_Caj_15 and Q_Caj17 samples (Sandoval et al. 2013). As reviewer suggested, we commented about this with following phrases in lines 286-291:

“Probably due to the absence of published Y-STR data on contemporary northern coastal Peruvian populations, Salas et al. [8] did not detect the genetic relationship with the child mummy, as did other works using autosomal aDNA studies [11,12]. However, the inclusion of the Cajamarca population data (Q_Caj15 and Q_Caj17 samples from Sandoval et al. [15]) could have changed their interpretations. These studies show us how, in some cases, a lack of population data may lead to different conclusions or interpretations.”

If my minor comments and suggestions are addressed, I am inclined to recommend the publication of this paper.

Thank you.

Reviewer 2 Report

I would like to congratulate the authors on the relevance of the study. Below I indicate some points that can be improved.

Line 28: Why is the word “child” in capital letters?

Lines 92-94: To investigate the potential genetic links of the child mummy’s paternal lineage, we 92 conducted a study comparing the Y-chromosome profile of the child mummy with that 93 of modern Andean and northern Peruvian coastal populations.

The aim of the work seems to lead to error. It seems the authors would analyse the mummy again, when in fact they will analyze modern people and then compare with the results attributed to the mummy.

 Lines 159-160: To investigate whether the Aconcagua child mummy belonged to a previously undescribed Y-chromosome subhaplogroup with other individuals from the northern Peruvian coast

The authors intend to check whether or not the mummy could belong to a new branch of the Y chromosome. Shouldn't this be specified at the beginning as another objective (lines 92-94)? since they intend to review the previously determined lineage?

Lines 265-267: It repeats an idea largely explained in the introduction.

Lines 289-293: It was already said on lines 40-44.

Line 330: “because of it appears to be a case (…)” This sentence is not well-written

Despite being an interesting study, the way it is written does not allow for fluid reading. In each paragraph the authors get lost in the Historical part, including too many commas in the text, meaning that, in the end, it is not easy to retain much of the information described. For example, the sentence that starts on line 319 only ends on line 325. A lot of information comes together that, for those who read the article, is difficult to understand.

Minor editing of the English language required.

Author Response

I would like to congratulate the authors on the relevance of the study. Below I indicate some points that can be improved.

Line 28: Why is the word “child” in capital letters?

The typo error was fixed.

Lines 92-94To investigate the potential genetic links of the child mummy’s paternal lineage, we 92 conducted a study comparing the Y-chromosome profile of the child mummy with that 93 of modern Andean and northern Peruvian coastal populations.

The aim of the work seems to lead to error. It seems the authors would analyse the mummy again, when in fact they will analyze modern people and then compare with the results attributed to the mummy.

We rephrased:

“To investigate the potential genetic links of the child mummy’s paternal lineage, we conducted a study searching the Peruvian Y-chromosome database (contemporary Andean and northern Peruvian coastal populations) and compared it with the genetic profile of the child mummy [3,8]. Thus, to answer this genealogical question, we utilized SNPs and STRs from the non-recombining region of the Y-chromosome. We found consistency with previous autosomal ancient DNA studies.”

 Lines 159-160To investigate whether the Aconcagua child mummy belonged to a previously undescribed Y-chromosome subhaplogroup with other individuals from the northern Peruvian coast

The authors intend to check whether or not the mummy could belong to a new branch of the Y chromosome. Shouldn't this be specified at the beginning as another objective (lines 92-94)? since they intend to review the previously determined lineage?

To check was not the aim. We only intended to see if there was a specific branch (sublineage of Q-M3 haplogroup) for the child genetic profile, but it allows for no finer resolution due to lack of more data. We rephrased it:

“To refine whether the Aconcagua child mummy belonged to a previously undescribed Y-chromosome subhaplogroup with other individuals from the northern Peruvian coast, we compared it with the data from four ancient DNA reports by Bongers et al. [12] and Nakatsuka et al. [11] with sufficient coverage.”

Lines 265-267: It repeats an idea largely explained in the introduction.

In Discussion the idea was retaken to add other specific places (volcanoes) or similar burial findings.

Lines 289-293: It was already said on lines 40-44.

Observation of the peculiarities of the Capacocha ritual that involved the “Inka” child mummy, have led some scholars to suggest a close association with the pre-Columbian Peruvian coastal cultures. The ritual includes a cotton “cumbi” with pelican and sea wave iconographies, as well as objects made from Spondylus shells [2,4].

We deleted repetitive ideas:

“After observing the peculiarities of the capacocha ritual that involved the Inka child mummy, some scholars were led to suggest a close association with the pre-Columbian Peruvian coastal cultures [2,4].”

Line 330: “because of it appears to be a case (…)” This sentence is not well-written

This was rephrased:

“Also, the haplotype of the CHP-352 individual was very differentiated from Jivaroan-speaking people. This self-identified individual was likely sampled in the Chachapoyas city [35], and as an outlier, this Jivaroan was not included in the analyses. However, it could be interpreted as an occurrence of gene flow from the north Coast/Andes to a Jivaroan community (northwestern Amazonia, Amazonas department). Nevertheless, there might be another interpretation.”

Despite being an interesting study, the way it is written does not allow for fluid reading. In each paragraph the authors get lost in the Historical part, including too many commas in the text, meaning that, in the end, it is not easy to retain much of the information described. For example, the sentence that starts on line 319 only ends on line 325. A lot of information comes together that, for those who read the article, is difficult to understand.

Minor editing of the English language required.

The manuscript was sent to Scribendi Inc for the revision and edition of English.

Thank you.

Round 2

Reviewer 2 Report

With these modifications, I think the article is ready to be published. It's much more "understandable".

The authors mention that the article was sent for English review, so I have nothing more to say.